# CYP1A1, VEGFA and Adipokine Responses of Human Adipocytes Co-exposed to PCB126 and Hypoxia

**DOI:** 10.3390/cells11152282

**Published:** 2022-07-24

**Authors:** Zeinab El Amine, Jean-François Mauger, Pascal Imbeault

**Affiliations:** 1School of Human Kinetics, Faculty of Health Sciences, University of Ottawa, Ottawa, ON K1N 6N5, Canada; zelam100@uottawa.ca (Z.E.A.); jmauger@uottawa.ca (J.-F.M.); 2Institut du Savoir Montfort, Hôpital Montfort, Ottawa, ON K1K 0T2, Canada

**Keywords:** hypoxia, polychlorinated biphenyl, inflammatory adipokines, human adipocytes, AhR, ARNT

## Abstract

It is increasingly recognized that hypoxia may develop in adipose tissue as its mass expands. Adipose tissue is also the main reservoir of lipophilic pollutants, including polychlorinated biphenyls (PCBs). Both hypoxia and PCBs have been shown to alter adipose tissue functions. The signaling pathways induced by hypoxia and pollutants may crosstalk, as they share a common transcription factor: aryl hydrocarbon receptor nuclear translocator (ARNT). Whether hypoxia and PCBs crosstalk and affect adipokine secretion in human adipocytes remains to be explored. Using primary human adipocytes acutely co-exposed to different levels of hypoxia (24 h) and PCB126 (48 h), we observed that hypoxia significantly inhibits the PCB126 induction of cytochrome P450 (CYP1A1) transcription in a dose-response manner, and that Acriflavine (ACF)—an HIF1α inhibitor—partially restores the PCB126 induction of CYP1A1 under hypoxia. On the other hand, exposure to PCB126 did not affect the transcription of the vascular endothelial growth factor-A (VEGFA) under hypoxia. Exposure to hypoxia increased leptin and interleukin-6 (IL-6), and decreased adiponectin levels dose-dependently, while PCB126 increased IL-6 and IL-8 secretion in a dose-dependent manner. Co-exposure to PCB126 and hypoxia did not alter the adipokine secretion pattern observed under hypoxia and PCB126 exposure alone. In conclusion, our results indicate that (1) hypoxia inhibits PCB126-induced CYP1A1 expression at least partly through ARNT-dependent means, suggesting that hypoxia could affect PCB metabolism and toxicity in adipose tissue, and (2) hypoxia and PCB126 affect leptin, adiponectin, IL-6 and IL-8 secretion differently, with no apparent crosstalk between the two factors.

## 1. Introduction 

There is accumulating evidence that as adipose tissue mass expands, insufficient angiogenesis and cellular hypertrophy may limit oxygen availability to adipocytes, a concept that has been called ‘adipose tissue hypoxia’ [1,2]. Studies conducted in rodent models have shown that AT oxygenation is lower in rodent models with obesity compared to their lean counterparts [3,4,5]. Lower adipose tissue oxygenation in the context of excess adiposity has also been confirmed in several human studies [6,7,8,9,10,11] but not all of them [12]. On the other hand, many in vitro studies have shown that hypoxia negatively alters the lipid metabolism in adipocytes, mainly by decreasing their lipogenic capacity and increasing their basal lipolytic rate [13,14,15].

The cellular response to hypoxia is mainly induced through the hypoxia-dependent stabilization of the transcriptional factor hypoxia-inducible factor-1α subunit (HIF1α). Stabilized HIF1α dimerizes with its cofactor, aryl hydrocarbon receptor translocator (ARNT), and the complex induces the transcription of several genes containing hypoxia response elements (HRE) [16,17]. Many genes expressed in adipose tissue have been reported to contain HRE, including leptin [18], vascular endothelial growth factor-A (VEGFA) [17], glucose transporter 1, GLUT1 [19], and interleukin-6 (IL-6) [4,20,21], implying that hypoxia affects critical adipose tissue features such as inflammation, insulin resistance, glucose intolerance, and angiogenesis [4,15,22,23,24]. Moreover, the expression and release of adipocytes’ signature hormones, leptin and adiponectin, are remarkably altered during obesity, an effect that has been shown to be HIF1α-dependent. The fact that the alterations induced by hypoxia in the adipose tissue mirror the adipose tissue alterations observed in the context of obesity strongly suggests that adipose tissue hypoxia could play an important role in the development of the health complications associated with obesity [25,26]

Adiposity levels have also been strongly correlated with the accumulation of lipophilic persistent organic pollutants (POPs) [27,28,29,30] such as polychlorinated biphenyls (PCBs), which in turn have been correlated with many of the complications linked to obesity [31,32,33,34]. Among PCBs, coplanar dioxin-like congeners such as PCB77 and PCB126 are among the most toxic and widespread in the environment [35]. While the classical cellular response to dioxin-like compound exposure consists of an increased expression of detoxifying cytochrome enzymes, namely cytochrome P450 (an enzyme regulated by the CYP1A1 gene) [36], these pollutants have also been reported to alter adipose tissue metabolism and endocrine function. As such, PCB126 exposure has been linked to the increased secretion of leptin and pro-inflammatory cytokines such as interleukin—IL-6 and IL-8—and the decreased secretion of adiponectin [32,37,38,39,40,41,42,43]. Most of these effects are thought to arise from the binding of PCBs to the aryl hydrocarbon receptor (AhR), a transcription factor that modulates the expression of genes bearing dioxin- and xenobiotic-response elements (DRE/XRE). Interestingly, the genomic action of AhR requires its dimerization with ARNT, the same co-factor involved in the hypoxia signaling pathway. This dual role of ARNT is best illustrated by the fact that conditional ARNT knockout mice are not able to induce AhR target genes in response to TCDD exposure, nor HIF1α target genes under hypoxic-like conditions [44].

Hence, ARNT is required as a common co-factor for these two signaling pathways, which can lead to crosstalk and interference. The crosstalk between the hypoxia-induced and the xenobiotic-induced pathways has been reviewed in detail by Vorrinck and Domann [45]; it appears that, in most cases, co-exposure to dioxin-like POPs and hypoxia results in the induction of the hypoxia response and a repression of the xenobiotic response [46]. However, this crosstalk has been almost exclusively studied in liver cell lines. This is likely to be because the liver is a highly metabolically active organ in which hypoxia may develop, and it is also the main organ responsible for xenobiotic detoxification and excretion. However, because excessive adiposity can give rise to adipose tissue hypoxia and lipophilic xenobiotics accumulation, this crosstalk is also most likely to occur in adipocytes, especially in individuals with obesity.

The aim of the present study was to characterize the interaction between hypoxia and dioxin-like PCB126 exposure on the expression of genes induced by hypoxia (VEGFA) and dioxin (CYP1A1), as well as on the secretion of IL-6, IL-8, leptin and adiponectin in human adipocytes.

## 2. Methods and Materials

### 2.1. Human Subcutaneous Preadipocyte Culture

Human subcutaneous preadipocytes and complete media were obtained commercially (ZenBio Inc., Research Triangle Park, NC, USA), and were used according to the manufacturer’s instructions. Preadipocytes were plated in Preadipocyte Medium (PM-1) at a density of 40,000 cells/cm^2^ in 24-well plates (Corning Life Science, Corning, NY, USA); these were incubated in a sterile, humidified incubator at 37 °C with 5% CO_2_ until the cells reached full confluence (24–48 h). PM-1 was then replaced with Differentiation Medium (DM-2) to induce differentiation. Seven days post-induction, the cells were fed using Adipocyte Maintenance Medium (AM-1). Then, 14 days post-induction, the cells displayed multiple lipid droplets; these were considered mature adipocytes, and treatments were initiated.

### 2.2. Human Adipocytes’ Exposure to PCB126 and Hypoxia

A 10 mM stock solution of 3,3’,4,4’,5-pentachlorobiphenyl (PCB126, Ultra Scientific, North Kingstown, RI, USA) was prepared with sterile-filtered DMSO. Mature adipocytes were exposed to either DMSO (control), 1 µM PCB126 or 10 µM PCB126 in AM-1 medium for 24 h. The control and PCB126 treatments were prolonged for another 24 h under either 21%, 10% or 3% O_2_. Cell culturing has traditionally been conducted under 21% O_2_; as such, this O_2_ concentration was chosen as a control condition. We used 3 and 10% O_2_ because these cover the physiological O_2_ tension range observed in the human adipose tissue of individuals with different adiposity levels [6,7,8,9,10,11,12]. Hypoxia treatments were conducted using a HERA cell 150iO_2_ incubator (Thermo Fisher Scientific, Waltham, MA, USA) and medical nitrogen. The cells were therefore exposed to PCB126 for a total of 48 h, and to hypoxia for 24 h. Acriflavine (ACF, Millipore Sigma, Oakville, ON, Canada), an HIF1α-inhibitor that binds directly to HIF1α and HIF2α and inhibits HIF1α dimerization without affecting cell proliferation or survival [47], was prepared in sterile PBS and added (at a final concentration of 5 µM) 4 h prior to the 24-h hypoxia/normoxia exposure. After the 48-h treatment phase the media were collected, and the cells were rapidly washed with sterile ice-cold PBS and kept on ice until further processing. Each experiment was repeated 3 times (*n* = 3), with all treatments being run in triplicate each time.

### 2.3. Adipokine Quantification

The media were stored at −80 °C until they were analyzed. The adipokine concentrations were determined by commercially available colorimetric ELISAs (R&D Systems, Minneapolis, MN, USA). The assays were performed following the manufacturer’s protocols. According to the manufacturer, the assays’ sensitivities were: leptin, 7.8 pg/mL; IL-6, 0.09 pg/mL; and adiponectin, 0.9 ng/mL, IL-8 7.5 pg/mL. In order to avoid freeze–thaw cycles, all of the adipokines were measured on the same day for samples from the same experiment. The measurements were carried out in singlets.

### 2.4. RNA Extraction and Quantification by qPCR

Immediately after washing, the cells were lysed in RLT buffer (Qiagen, Germantown, MD, USA) supplemented with B-mercaptoethanol and shredded using Qiagen QiaShredder columns. The lysates were then kept at −80 °C until they were processed further. Total RNA extraction was performed using Qiagen’s RNeasy kit, following the manufacturer’s protocol. The extracted RNA was reverse transcribed using Qiagen’s QuantiTect Reverse Transcription kit, and real-time polymerase chain reactions (rtPCR) were performed for VEGFA, CYP1A1 and B-actin using 20 ng cDNA, Qiagen’s QuantiTect Primers and Montreal Biotech’s (Dorval, QC, Canada) MBI EVOlution EvaGreen qPCR master mix on a RotorGene (Corbett Research, Australia) thermocycler. The Ct values for VEGFA and CYP1A1 were normalized to β-actin, and the ΔΔCt method was used to calculate fold changes in gene expression for each experimental condition using the DMSO-21% O_2_ exposure as the reference treatment.

### 2.5. Statistical Analysis

All data are expressed as a mean ± standard error. Three-way ANOVAs were conducted with the PCB126 concentration, oxygen concentration, and ACF as fixed factors for all the variables. For gene expression, statistical analyses were performed on the ΔCt values. Post hoc pairwise comparisons between levels of significant main factors or interaction terms were conducted using the Tukey HSD test. Significance was set at *p* < 0.05. All of the statistical analyses were performed using JAMOVI statistical software version 2.2.5 (Jamovi project, Sydney, Australia).

## 3. Results

### 3.1. Effects of Hypoxia and PCB126 on CYP1A1 and VEGFA Expression 

PCB126 exposure induced CYP1A1 expression in a comparable and dose-dependent fashion under normoxia and 10% O_2_ (Figure 1). This effect was still present but greatly attenuated under 3% O_2_ (the main effect of O_2_ *p* < 0.001). ACF increased the expression of CYP1A1, especially when combined with 10 µM PCB126 (ACF × PCB126 interaction *p* = 0.012) and restored the CYP1A1 response to PCB126 at 3% O_2_ to levels comparable to that observed under normoxia and 10% O_2_ without ACF. 

As a reference point for O_2_ tension exposure, VEGFA gene expression was measured. VEGFA expression was significantly increased in a dose-dependent manner by hypoxia (the main effect of O_2_ *p* < 0.001) (Figure 2). PCB126 exposure did not affect this response (the main effect of PCB126 *p* = 0.476). The increase in VEGFA expression under hypoxia was completely abolished by HIF1α inhibition with ACF (O_2_ × ACF interaction *p* < 0.001). 

### 3.2. Effects of Hypoxia and PCB126 on Adipokine Secretion 

Leptin secretion was significantly increased under hypoxia in a dose-dependent fashion (roughly 300% and 500% at 10% and 3% O_2_, respectively, with a main effect of O_2_ *p* < 0.0001) (Figure 3). This hypoxia-induced effect on leptin levels was not affected by PCB126 exposure. HIF1α inhibition with ACF completely abolished the stimulating effect of hypoxia on leptin secretion (O_2_ × ACF interaction *p* < 0.001). 

The adiponectin concentrations were significantly decreased under hypoxia, but only in the most severe hypoxia condition (−25% at 3% O_2_, the main effect of hypoxia *p* < 0.0001). HIF1α inhibition with ACF partially but not completely restored adiponectin secretion under 3% O_2_ (the main effect ACF *p* = 0.074) (Figure 4). Adiponectin secretion was not affected by PCB126 exposure. 

IL-6 secretion was increased by hypoxia in a dose-dependent fashion (Figure 5) (the main effect of O_2_ *p* = 0.002). This effect was not repressed by ACF; rather, it was increased by 35% in all conditions (the main effect of ACF *p* = 0.001). A dose-dependent trend was observed between PCB126 and IL-6 secretion (the main effect of PCB126 *p* = 0.068). 

IL-8 secretion was not significantly affected by hypoxia but was increased overall by 50% and 60% at 1 µM and 10 µM PCB126, respectively (the main effect PCB126 *p* = 0.003) (Figure 6). IL-8 concentrations were also significantly increased by 25% with ACF treatment (the main effect ACF *p* = 0.031).

## 4. Discussion

The main objective of the present study was to characterize the interaction between hypoxia and dioxin-like PCB126 exposure, two stressors to which the responses involve the heterodimerization partner ARNT, in primary human adipocytes. We found that exposure to PCB126 significantly increased CYP1A1 expression—a classical element of the dioxin response—in a dose-response manner, and this response was significantly repressed after 24 h of exposure to 3% O_2_. In contrast, exposure to hypoxia induced VEGFA expression, a classical element of the hypoxia response, but this response was not altered by PCB126 exposure. Together, these observations suggest that hypoxia interferes with the xenobiotic sensing pathway, and that highly potent dioxin-like compounds such as PCB126 do not interfere with the hypoxia response.

To our knowledge, this is the first study showing that CYP1A1 induction by PCB126 is significantly inhibited in human adipocytes exposed to hypoxia (Figure 1). This finding is in line with several other studies using different cell models co-exposed to dioxin/dioxin-like POP and hypoxia [48,49,50,51,52,53,54,55,56]. Still, some studies reported a downregulation of the hypoxia signaling pathway by POPs [57,58,59,60]; and in one instance, using aminoflavone, the inhibition was shown to be independent of AhR activation [60]. Furthermore, some studies even reported dioxin-like compounds and hypoxia to mutually inhibits each other pathways [45,61,62,63]. Most of these previous studies were almost exclusively conducted in human and rodent hepatic cell lines using TCDD [45,50,51,53,54,56,58,59] or benzo(a)pyrene [52,55,62] as an AhR ligand, and with different hypoxic modalities of various intensities and durations [46]. These experimental disparities in the body of work on POPs and hypoxia co-exposure make fine comparisons between studies almost impossible, but they emphasize the robustness with which hypoxia appears to interfere with the xenobiotic response. The present study extends our understanding of the xenobiotics–hypoxia interaction to human adipose tissue by showing that, as far as the traditional gene markers of hypoxia (VEGFA) and xenobiotic (CYP1A1) responses are concerned, acute co-exposure to hypoxia and PCB126 results in the full activation of the HIF1α signaling pathway and the downregulation of the AhR signaling pathway in human adipocytes.

Whether hypoxia interferes with the xenobiotic response solely because HIF1α stabilization prevents AhR dimerization with ARNT is, however, still unclear. The fact that ACF restored CYP1A1 induction by 10µM PCB126 under 3% O_2_ to levels comparable to normoxia and 10% O_2_ without ACF suggests so. However, because ACF significantly increased CYP1A1 expression in the presence of 10 µM PCB126 at both 21% and 10% O_2_, CYP1A1 expression with ACF treatment still appeared restrained at 10 µM PCB126 and 3% O_2_ (compared to normoxia and 10% O_2_ with ACF) (Figure 1). The incomplete inhibition of HIF1α by ACF appears unlikely because ACF treatment completely abolished VEGFA and leptin induction under hypoxia (Figure 2 and Figure 3). Therefore, our observations suggest that factors other than HIF1α stabilization may interfere with CYP1A1 induction by xenobiotics under severe hypoxia in human adipocytes. As such, it is noteworthy that ARNT has additional partners that may be activated under stress conditions and affect the AhR induced transcriptional response [64]. Further studies are warranted to confirm the presence and identity of such factors that may adversely affect adipocyte xenobiotic metabolism under hypoxia independently of HIF1α signaling.

### 4.1. Adipokine Secretion under Co-Exposure to Hypoxia and PCB126 

Because hypoxia and dioxin-like xenobiotics have been reported to affect the secretion of many adipokines, we sought to determine how co-exposure to hypoxia and PCB126 would affect leptin, IL-6, IL-8 and adiponectin secretion compared to exposure to hypoxia and PCB126 alone. We observed that leptin secretion was increased dose-dependently by hypoxia, an effect that was mitigated by ACF (Figure 3). These results are in line with previous evidence demonstrating that the transcriptional activity of the leptin gene is upregulated by the HIF1a-ARNT complex under hypoxia [18,65]. PCB126 exposure had no effect on leptin secretion per se and did not affect the leptin response to hypoxia. Contrary to the well-described effects of hypoxia on leptin secretion, whether PCB126 exposure or POP exposure in general affect AT leptin secretion is still obscure. In humans, one study reported a correlation between the plasma concentrations of some POPs and leptinemia, especially in women, but this relationship was not observed with any PCB congeners [66]. In rodents, exposure to PCB126 at doses from 0.1 to 10 μg/kg for 15 d was reported to have no effect on leptinemia [67] while in vitro, leptin secretion was not altered in 3T3-L1 cells exposed for 24 h to low doses of PCB126 (between 1 and 100 nM) [38]. Our observations, combined with previous reports, support the notions that acute PCB126 exposure does not seem to impact leptin secretion and that PCB126 does not affect the increase in leptin secretion under hypoxia, at least under the experimental conditions we tested.

IL-6 secretion increased dose-dependently in response to hypoxia (Figure 5), which is consistent with previous studies conducted in vitro on cultured human adipocytes [4,20,21]. IL-6 secretion was also increased dose-dependently by PCB126, with increases of 11% at 1 µM PCB126 and 34% at 10 µM PCB126 compared to DMSO, but this did not reach statistical significance (*p* = 0.068). Previous reports on the effect of PCB126 exposure on IL-6 secretion are conflicting. Gourronc et al. [40] observed a twofold increase in IL-6 secretion in cultured adipocytes following a 48-h exposure to 10 µM PCB126. On the other hand, Loiola et al. [67] showed that a 15-day exposure to PCB126 up to 10 μg/kg decreased retroperitoneal adipose tissue IL-6 mRNA levels in rats, while Caron et al. [38] showed no effect of a 24-h exposure to low PCB126 doses up to 100 nM on IL-6 secretion in 3T3-L1. Interestingly, TCDD, the most potent dioxin, has been demonstrated to repress LPS-induced IL-6 production in bone marrow stromal cells, suggesting that the alleged binding of activated AhR to the IL-6 XRE may, in some cases, downregulate IL-6 expression [68]. As for our observations, they suggest that acute exposure to PCB126 and hypoxia may have additive effects on IL-6 secretion in human adipocytes, such that the combination of both PCB126 accumulation and hypertrophy-induced hypoxia may lead to greater inflammatory responses and metabolic disturbances in human adipose tissue than either stressor alone.

IL-8, another pro-inflammatory adipokine, behaved differently from leptin and IL-6 in that its secretion was induced by PCB126 exposure but was unaffected by hypoxia (Figure 6). The increase in IL-8 secretion by PCB126 is consistent with previous studies conducted in cultured adipocytes [40,69]. Interestingly, there is evidence that both IL-6 and IL-8 inductions by PCB126 and TCDD are dependent on AhR signaling [40,70,71,72]; however, neither adipokine appeared repressed under hypoxia in our experiments. These observations suggest that the PCB126 induction of IL-6 and IL-8 by AhR may be independent of AhR/ARNT dimerization in adipocytes. In this regard, it is increasingly well recognised that AhR can activate signalling pathways independently of ARNT dimerization (a non-classical mechanism of action); one such pathway is NF-κB [36,73], an important inflammation modulator and a regulator of IL-6 and likely IL-8 expression [74]. Based on our observations, it is most likely that the increases in IL-6 and IL-8 secretion by PCB126 even under hypoxia could occur through such an ARNT-independent AhR signalling pathway, as previously demonstrated [75,76].

An interesting observation regarding IL-6 and IL-8 is that ACF led to an overall 40% increase in IL-6 secretion, and to a 25% increase in IL-8 secretion. This effect of ACF was unexpected and cannot be easily explained. To our knowledge, the effect of ACF on inflammatory markers has never been studied per se, but some works suggest that ACF could have both anti-inflammatory [77] and inflammatory (DNA-disrupting) actions [78]. Although they are unexplained, these observations strongly support the fact that the robust increase in IL-6 secretion under hypoxia is not likely to be a result of HIF1α activation.

We observed that adiponectin secretion was reduced under 3% O_2_ but was not affected by PCB126 exposure (Figure 4). The reduction in adiponectin secretion under hypoxia is consistent with previous studies [4,21,65,79]. in line with our observations, Jiang et al. reported that 5 µM ACF restores adiponectin secretion in vitro in 3T3-L1 exposed to the chemical hypoxia mimetic CoCl_2_, and also demonstrated that HIF1α is responsible for the decrease in adiponectin secretion under hypoxia by affecting SOCS3-STAT3 signaling [79]. Interestingly, they reported—similarly to our findings—that 5 µM ACF does not fully restore adiponectin secretion in 3T3-L1 under chemical hypoxia. This suggests that although HIF1α may be an important regulator of adiponectin expression, other factors may limit adiponectin production in conditions in which HIF1α is activated. Contrary to hypoxia, PCB126 exposure had no effect on adiponectin secretion in our experiments. Previous studies of the effect of PCB126 on adiponectin secretion are conflicting. In vitro, Klingelhutz et al. [69] and Gadupudi et al. [80] both showed that exposing preadipocytes to 10 µM PCB126 during proliferation inhibits their differentiation into mature adipocytes and decreases adiponectin secretion up to 2 weeks after PCB126 treatment. On the other hand, Caron et al. [38] observed an increase in adiponectin secretion in insulin-sensitive 3T3-L1 treated for 24 h with PCB126 at doses of 10 and 100 nM. It has also been shown that a single PCB126 intraperitoneal injection (5 µmol/kg) in rats significantly increased serum adiponectin levels after 28 d [39]. Taken together, these observations are difficult to reconcile but may suggest that the effect of PCB126 on adiponectin may depend on the PCB dose as well as the timing and duration of exposure.

### 4.2. Limitations and Strengths 

The main limitations of the current study include the use of a single POP at supraphysiological doses, although these are commonly used, and the fact that both the hypoxia and PCB126 exposures were performed acutely. In living humans, adipocytes are exposed to a cocktail of different POPs as well as hypoxia for widely varying durations, which may exert different effects on adipocyte metabolism. Furthermore, in living organisms, adipocyte metabolism can possibly be altered indirectly through the effects exerted by POPs or hypoxia on other tissues. These limitations call for caution when extrapolating findings from our in vitro study to in vivo conditions. One strength of the current study lies on the fact that we used a range of O_2_ tensions: a commonly considered normoxic condition, although it is sometimes referred to as hyperoxia (21% O_2_); a mild hypoxic condition, though it is sometimes considered a physiological normoxic condition (10% O_2_); and a clearly hypoxic condition (3% O_2_) [81].

## 5. Conclusions

The present study supports the theories that hypoxia and xenobiotic signalling pathways interfere in human adipocytes, and that this interference is likely explained by HIF1α preventing the dimerization of AhR with ARNT, which may partially repress the xenobiotic transcriptional response. However, our results showed that PCB126 increased IL-6 and IL-8 secretion, an effect that is not repressed by hypoxia. This observation suggests that the PCB126 induction of IL-6 and IL-8 by AhR may be independent of AhR/ARNT dimerization, and that AhR can activate non-classical signaling pathways such as NF-kB. Therefore, the long-term co-exposure of adipocytes to hypoxia and POPs could accelerate the establishment of a pro-inflammatory state in the adipose tissue, a widely acknowledged crucial step in the development of obesity-related metabolic complications.

## Figures and Tables

**Figure 1 cells-11-02282-f001:**
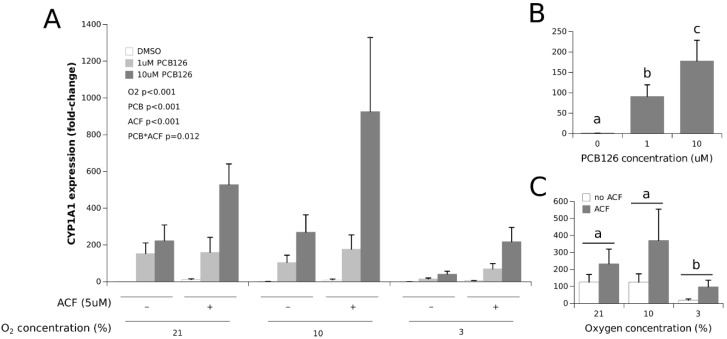
Effect of PCB126 and hypoxia on CYP1A1 expression in human adipocytes. Adipocytes were treated with DMSO (vector) or 1 and 10 µM PCB126 for 24 h and then subjected to 21, 10 or 3% O_2_ for another 24 h. Acriflavine (ACF) 5 µM was used as an HIF1α-inhibitor. Panel (**A**) illustrates the whole model. Markers of statistical difference for individual bars were omitted due to the complexity of the model. Panel (**B**) summarizes the main effect of PCB126 on CYP1A1 expression. Panel (**C**) summarizes the effects of hypoxia and ACF treatment on CYP1A1 expression. In panels B and C, bars or groups of bars not sharing a common letter are statistically different at *p* < 0.05. The letters in panel C refer to the effect of hypoxia alone, although the effect of ACF is also significant. Each bar represents the mean ± SEM of three separate experiments for each treatment group.

**Figure 2 cells-11-02282-f002:**
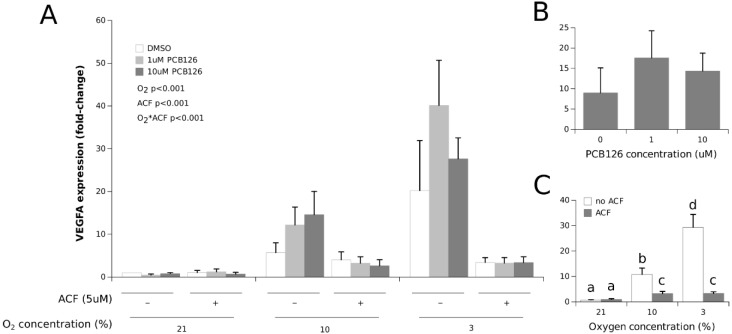
Effect of PCB126 and hypoxia on VEGFA expression in human adipocytes. Adipocytes were treated with DMSO (vector) or 1 and 10 µM PCB126 for 24 h, and then subjected to 21, 10 or 3% O_2_ for another 24 h. Acriflavine (ACF) 5 µM was used as the HIF1α-inhibitor. Panel (**A**) illustrates the whole model. Markers of statistical difference for individual bars were omitted due to the complexity of the model. Panel (**B**) summarizes the main effect of PCB126 on VEGFA expression. Panel (**C**) summarizes the effects of hypoxia and ACF treatment on VEGFA expression. In panel C, bars not sharing a common letter are statistically different at *p* < 0.05. Each bar represents the mean ± SEM of three separate experiments for each treatment group.

**Figure 3 cells-11-02282-f003:**
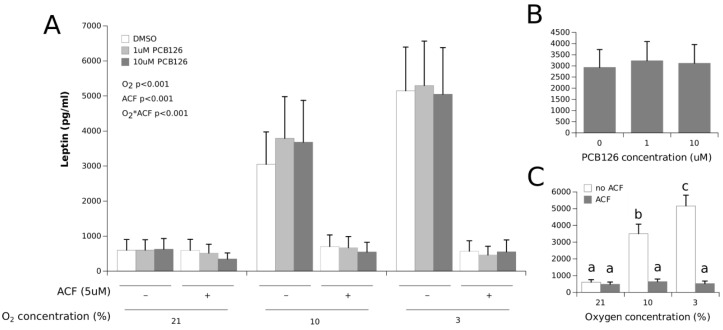
Effect of PCB126 and hypoxia on the media leptin concentration in human adipocytes. Adipocytes were treated with DMSO (vector) or 1 and 10 µM PCB126 for 24 h, and then subjected to 21, 10 or 3% O_2_ for another 24 h. Acriflavine (ACF) 5 µM was used as an HIF1α-inhibitor, then subjected to 21, 10 or 3% O_2_ for another 24 h. Acriflavine (ACF) 5 µM was used as an HIF1α-inhibitor. Panel (**A**) illustrates the whole model. Markers of statistical difference for individual bars were omitted due to the complexity of the model. Panel (**B**) summarizes the main effect of PCB126 on the media leptin concentrations. Panel (**C**) summarizes the effects of hypoxia and ACF treatment on the media leptin concentrations. Bars not sharing a common letter are statistically different at *p* < 0.05. Each bar represents the mean ± SEM of three separate experiments for each treatment group.

**Figure 4 cells-11-02282-f004:**
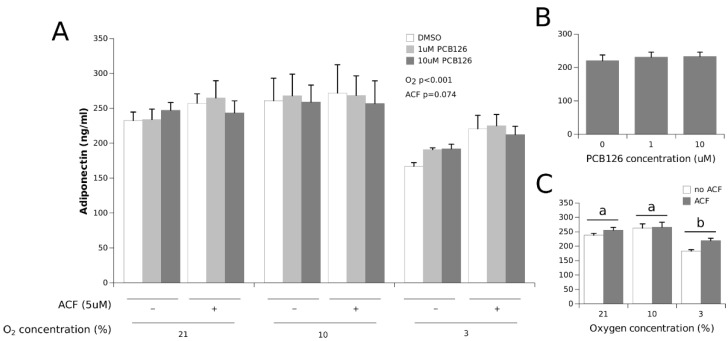
Effect of PCB126 and hypoxia on the media adiponectin concentration in human adipocytes. Adipocytes were treated with DMSO (vector) or 1 and 10 µM PCB126 for 24 h, then subjected to 21, 10 or 3% O_2_ for another 24 h. Acriflavine (ACF) 5 µM was used as an HIF1α-inhibitor. Panel (**A**) illustrates the whole model. Markers of statistical difference for individual bars were omitted due to the complexity of the model. Panel (**B**) summarizes the main effect of PCB126 on the media adiponectin concentrations. Panel (**C**) summarizes the effects of hypoxia and ACF treatment on the media adiponectin concentrations. Pairs of bars (representing the main effect of oxygen) not sharing a common letter are statistically different at *p* < 0.05. Each bar represents the mean ± SEM of three separate experiments for each treatment group.

**Figure 5 cells-11-02282-f005:**
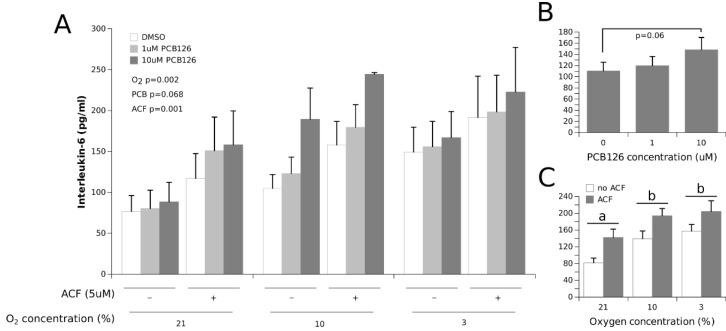
Effect of PCB126 and hypoxia on the media IL-6 concentration in human adipocytes. Adipocytes were treated with DMSO (vector) or 1 and 10 µM PCB126 for 24 h, then subjected to 21, 10 or 3% O_2_ for another 24 h. Acriflavine (ACF) 5 µM was used as an HIF1α-inhibitor. Panel (**A**) illustrates the whole model. Markers of statistical difference for individual bars were omitted due to the complexity of the model. Panel (**B**) summarizes the trend between 0 and 10 µM PCB126 on media IL-6 concentrations. Panel (**C**) summarizes the effects of hypoxia and ACF treatment on media IL-6 concentrations. In panel C, pairs of bars (representing the main effect of oxygen) not sharing a common letter are statistically different at *p* < 0.05. Each bar represents the mean ± SEM of three separate experiments for each treatment group.

**Figure 6 cells-11-02282-f006:**
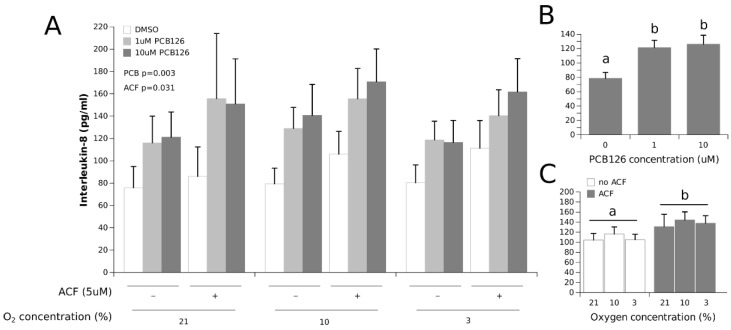
Effect of PCB126 and hypoxia on the media IL-8 concentration in human adipocytes. Adipocytes were treated with DMSO (vector) or 1 and 10 µM PCB126 for 24 h, then subjected to 21, 10 or 3% O_2_ for another 24 h. Acriflavine (ACF) 5 µM was used as an HIF1α-inhibitor. Panel (**A**) illustrates the whole model. Markers of statistical difference for individual bars were omitted due to the complexity of the model. Panel (**B**) summarizes the main effect of PCB126 on media IL-8 concentrations. Panel (**C**) summarizes the effects of hypoxia and ACF treatment on media IL-8 concentrations. In panels B and C, bars or groups of bars not sharing a common letter are statistically different at *p* < 0.05. Each bar represents the mean ± SEM of three separate experiments for each treatment group.

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
