# Peer review of "CYP1A1, VEGFA and Adipokine Responses of Human Adipocytes Co-exposed to PCB126 and Hypoxia"

_cells, 2022, doi:10.3390/cells11152282_

Round 1
Reviewer 1 Report
This study describes experiments on primary human preadipocytes co-exposed to hypoxia and CYP1A1 inductor PCB126. The authors found that hypoxia and xenobiotic signaling pathways interfere in human adipocytes and hypoxia repress the xenobiotic transcriptional response by preventing the dimerization of AhR with ARNT. But pollutants in fact are not silent in adipose tissue by activating non-classical signal transduction pathways of the AhR. They conclude that co-exposure of adipocyte to hypoxia and POPs could accelerate the establishment of a pro-inflammatory state in the adipose tissue.
The work appears to be interesting, important and well presented.
Minor revisions
1. I recommend to improve clarity: add to the conclusion that PCB126 induction of IL-6 and IL-8 by AhR may be independent of AhR/ARNT dimerization and AhR can activate non-classical signaling pathways - in this case NF-kB
2. In the figures it would be good to put asterisks indicate statistical significance. In its current form, the results are difficult to understand
Author Response
Please see the attachment file.
N.B. kindly see any amendments or addition in red font in the revised manuscript.

Reviewer 2 Report
This is a solid, well-conducted study that examines a question of interest. The results are clear and convincing, and the manuscript is well-written. The authors are requested to consider the following specific points:
1. I am not sure that it is appropriate to describe the cells as ‘human differentiated preadipocytes’ in that this implies that they were actually preadipocytes whereas the experiments were performed in effect on adipocytes - which had been differentiated from preadipocytes. Better to describe them as ‘adipocytes differentiated from preadipocytes’?
2 2. The use of 21, 10 and 3% oxygen is a good approach to examining the effects of hypoxia, but it would be helpful to add a sentence in the Introduction or Materials and Methods explaining the strategy/choices.
3 3. VEGF was examined at the level of gene expression, whereas leptin and the other adipokines were assessed as the secreted protein. Ideally, both mRNA and secreted protein would be measured. A comment on the rationale for the different approaches should be given. Do you have data for both mRNA and protein?
4. Discussion. It is inappropriate to say that ‘IL-6 secretion also TENDED to be increased…’ If you use statistics and there is no statistically significant difference then you should not imply that there is actually a difference!.
Author Response
please see attachment file.
N.B. kindly note that any amendments or addition is added in red font in the original manuscript.
